# A Pilot Analysis for a Multicentric, Retrospective Study on Biodiversity and Difficult-to-Treat Pathogens in Burn Centers across the United States (MICROBE)

**DOI:** 10.3390/pathogens13080628

**Published:** 2024-07-27

**Authors:** Lindey C. Lane, David M. Hill

**Affiliations:** Department of Pharmacy, Regional One Health, Memphis, TN 38103, USA; dmhill@regionalonehealth.org

**Keywords:** burn, antimicrobial resistance, difficult to treat pathogens, risk factors

## Abstract

Following burn injury, patients are at increased risk of infection and are often cited as having a high incidence of difficult-to-treat pathogens (DTp). The purpose of this study is to determine the incidence of DTp after burn injury, which factors are associated with their development, and subsequent outcomes. This single-center, retrospective study assessed patients with thermal or inhalation injury who had a positive culture resulting in initiation of treatment (i.e., excision, topical, or systemic antimicrobials). Demographic data, pathogen and resistance profiles, and prior exposure to topical and systemic antimicrobials were collected. Pathogens were considered DTp if they were multi-drug-resistant (MDR), extensively drug-resistant (XDR), methicillin-resistant *Staphylococcus aureus* (MRSA), extended-spectrum beta-lactamase (ESBL)-producing, AmpC-producing, carbapenem-resistant, difficult-to-treat resistance (DTR) *Pseudomonas* sp., carbapenem-resistant *Acinetobacter baumannii* (CRAB), or *Stenotrophomonas* spp. Sixty-five patients who grew 376 pathogens were included in the final analysis. Two-hundred thirteen (56.7%) pathogens were considered DTp. Prior exposure to 7 of the 11 collected topical antimicrobials and 9 of 11 systemic antimicrobial classes were significantly associated with future development of a DTp. This remained true for six and eight, respectively, after controlling for significant covariates via logistic regression. As there were only four deaths, a Cox-proportional hazard analysis was not feasible. The Kaplan–Meier plot according to DTp revealed a clear divergence in mortality (Log rank *p* = 0.0583). In this analysis, exposure to topical and systemic antibiotics was associated with the development of DTp. The results from this pilot study will inform the next iteration of multicenter study.

## 1. Introduction

The prevalence of antimicrobial-resistant pathogens is increasing and is of major concern to public health. The *CDC*’s *2019 Antimicrobial Resistance Threat Report* indicated that more than 2.8 million infections are caused by drug-resistant pathogens, leading to the deaths of over 35,000 Americans each year [1]. Following burn injury, patients are at increased risk of infection due to the loss of their primary barrier to infection, the skin, and are often subject to protracted hospital stays and several courses of antimicrobial agents [2,3]. Additionally, there are studies reporting an increased incidence of difficult-to-treat pathogens (DTp) observed in patients treated for burn injuries, as compared to general hospitalized patients [4,5,6]. Gram-negative pathogens identified by the Infectious Disease Society of America as difficult-to-treat pathogens include: ESBL-producing enterobacterales, AmpC-producing enterobacterales, carbapenem-resistant enterobacterales, multidrug-resistant (MDR) *Pseudomonas aeruginosa*, carbapenem-resistant *Acinetobacter baumannii* (CRAB), and *Stenotrophomonas maltophilia* [7]. Infections due to DTp lead to the development of severe, often fatal, infections due to lack of sensitivity to many current antimicrobial treatments [8,9,10].

The development of infections due to DTp is likely multifactorial. Selection pressure from systemic antibiotic therapy has been shown to play a role in the development of antibiotic resistance, particularly multiple exposures to empiric antibiotic therapy [11,12,13,14,15,16]. In fact, a study evaluating the impact of antibiotic exposure on the development of MDR ventilator-acquired pneumonia in trauma patients identified limiting prophylactic antibiotic days as the only modifiable risk factor [17]. Topical antimicrobial therapy, often utilized in burn injury, has also been shown to be associated with the development of MDR organisms [18,19], as well as impaired wound healing [20]. Studies illustrate the importance of early burn wound excision and definitive closure on the development of invasive wound infections and sepsis, though defining these timeframes still requires further elucidation [21,22,23]. Necrotic tissue is a rich medium for pathogen growth and will propagate an uncontrolled inflammatory response, leading to high rates of sepsis and increased exposure to antimicrobials, highlighting the importance of prompt source control [21,24].

Many studies have evaluated the diversity of the microbiological ecosystem and antibiotic sensitivity of burn centers [25,26,27]. However, none have sufficiently explored the multifaceted landscape to determine factors that may pose the greatest risk of developing DTp. The development of highly resistant organisms is associated with increased mortality in hospitalized patients [28,29,30]. Currently, studies of the impact of MDR organisms on mortality in burn patients have been unable to detect a significant increase in mortality, likely due to methodology and small sample size [31,32]. Sepsis is the primary cause of mortality in burn patients who survive the initial resuscitation [33]. Prevention is paramount. Therefore, priority should be placed on identifying risk factors associated with the development of infections with DTp. As mentioned, due to the variance in practice between centers and reported pathogens, it is essential to perform this study utilizing multicenter data. The primary objective of this pilot study is to evaluate the incidence of DTp as well as identify risk factors and treatment outcomes associated with their development in order to better inform the next iteration (i.e., multicenter study).

## 2. Materials and Methods

### 2.1. Study Design and Patient Population

The study was approved by two institutional review boards (IRB), the University of Tennessee Health Science Center and the Regional One Health Research Institute (23-09578-XP). Both review boards waived the necessity of informed consent. This study was a single-center retrospective study of patients admitted to an American Burn Association-verified burn center between 1 January 2021 and 31 December 2021. Patients were screened for inclusion if they had a positive culture during the index hospitalization that led to the initiation of treatment (i.e., topical or systemic antimicrobials or burn wound excision). Patients were excluded for any of the following: (1) pregnant, (2) less than 18 years of age, (3) incarceration, (4) they were not expected to survive (i.e., death or initiation of comfort measures within 7 days), (5) admission to the unit for non-burn injuries, or (6) subsequent admissions.

### 2.2. Data Collection

Data were collected manually from the electronic medical record during individual chart review and compared between those who did and did not develop an infection with a DTp. Demographic data collected included: age, race, sex, comorbidities, and the presence of hospital-acquired infection risk factors (HAI RF) on admission (i.e., intravenous access, history of chemotherapy, positive urine drug screen or reported social history, resident in a nursing home or long-term acute care hospital, admission to the hospital in the last 90 days, or end-stage renal disease requiring dialysis). The collected burn injury characteristics included: mechanism of injury, % total body surface area burned (TBSA), % partial and full thickness injury, and presence and grade of inhalation injury. Patient outcome data included: total length of stay (if patient survived index hospitalization) or days to mortality. The surgical course was noted with data collected on number of surgical interventions, days to first burn wound excision, and days to final grafting procedure.

Culture results were collected, and susceptibility results were analyzed. Organisms in this study could have been isolated from any source including skin swabs, tissue cultures following surgical excision, bronchoalveolar lavage, sputum, tracheal aspirate, urine, bone, or blood. Due to the retrospective nature of the study and resource limitations, specific genetic data for resistance mechanisms are not routine at the study center and were not available for classification. Organisms were considered to be DTp if they fell into one or more of the following categories: MDR (non-susceptibility to at least one agent in three or more antimicrobial categories), XDR (non-susceptibility to at least one agent in all but two or fewer antimicrobial categories), vancomycin-resistant, confirmed or presumed ESBL-production [ceftriaxone minimum inhibitory concentration (MIC) ≥ 2 μg/mL], confirmed or presumed AmpC-production (resistance to ceftriaxone, cefotaxime, and ceftazidime), carbapenem-resistant (confirmed carbapenemase producers or resistant to at least one carbapenem), or methicillin-resistant *Staphylococcus aureus* (MRSA), CRAB, *Stenotrophomonas* spp., or DTR-*Pseudomonas* (not susceptible to at least one antibiotic in at least three antibiotic classes for which P. aeruginosa susceptibility is generally expected: penicillins, cephalosporins, fluoroquinolones, aminoglycosides, and carbapenems). Susceptible pathogens were defined as strains whose MIC were interpreted to be susceptible to a given antibiotic. Non-susceptible pathogens were defined as strains whose MICs were interpreted to be resistant or intermediate to a given antibiotic. An organism’s intrinsic resistance was taken into consideration and the organism was not classified as resistant if intrinsically resistant (e.g., *Enterococcus casseliflavus* was not considered to be vancomycin-resistant). Organisms not meeting any of the above criteria were classified as non-DTp. Information on culture sources and length of stay prior to culture obtainment were also collected.

Hierarchical data were collected according to patient, pathogen, and date of infection. Exposures of topical and systemic antimicrobials were tracked over time. Exposure was recorded at the pathogen level and was defined as exposure to any patient site (i.e., exposure at the patient level) prior to each culture obtainment. For example, a single patient could have had an infection present early in the hospital stay that may have had different (likely less) exposures than pathogens that were treated a month later in the same stay. Topical agents analyzed in this study included: mafenide, bacitracin, silver sulfadiazine, mupirocin, Dakin’s, silver nitrate solution, hypochlorous acid, other topicals, solid silver dressings, as well as chlorohexidine gluconate (CHG) and nasal decolonization. Exposures to systemic antimicrobial classes were also collected, including: non-pseudomonal beta-lactams (i.e., cefazolin, ceftriaxone, ampicillin, or ampicillin-sulbactam), anti-pseudomonal beta-lactams (i.e., piperacillin-tazobactam, cefepime), carbapenems (i.e., meropenem, ertapenem, or imipenem-cilastain), fluoroquinolones (i.e., ciprofloxacin or levofloxacin), anti-MRSA agents (i.e., vancomycin, daptomycin, or linezolid), aminoglycosides (i.e., amikacin, gentamicin, or tobramycin), extended-spectrum beta-lactam beta-lactamase inhibitors (i.e., ceftazidime-avibactam, meropenem-vaborbactam, or ceftolozane-tazobactam), metronidazole, tetracyclines (i.e., doxycycline or minocycline), sulfamethoxazole-trimethoprim, and antifungals (i.e., fluconazole, micafungin, or isavuconazole).

During this study period, the institution’s microbiology laboratory utilized the Brucker MALDI Biotyper for the identification of bacteria and yeast. This system uses mass spectrometry to determine the proteomic fingerprints of microorganisms and compares these to the research-use-only database for microbial identification. The BD Phoenix™ automated system was used as a backup method for identification, and was the primary system used for antimicrobial susceptibility testing. The identification of microbes was based on the results of 45 chromogenic and fluorogenic substrates. ESBL production identified in isolates of *E. coli*, *K. pneumoniae*, and *K. oxytoca* was based on differential responses to third-generation cephalosporins in the presence and absence of the beta-lactamase inhibitor clavulanic acid. Carbapenem resistance for other organisms was determined by resistance to either meropenem or ertapenem, which were the representative carbapenems on the antimicrobial susceptibility testing panel. The BD Phoenix system uses a Carbapenemase-producing Organism Detect Panel. Specific genetic testing for resistance at our institution requires samples to be sent out to consulting laboratories, and must be requested. The rules for antibiotic reporting and interpretation for MIC values from the BD Phoenix™ system were based on United States Food and Drug Administration-cleared interpretations built within the automated system. Additional or reflex-sensitivity testing that may have been reported was done iteratively or collaboratively according to intramural antimicrobial stewardship guidance and adhered to Clinical & Laboratory Standards Institute breakpoints using the Kirby–Bauer disk diffusion method. Disk diffusion was rare and utilized only in cases of ceftazidime-avibactam, meropenem-vaborbactam, or ceftolozane-tazobactam susceptibilities for multidrug-resistant *Acinetobacter* spp., *Enterobacterales* spp., or *Pseudomonas* spp. Quality control for Phoenix identification and MICs followed the package insert for the Phoenix products. Kirby–Bauer interpretation, reporting and quality control followed CLSI recommendations.

### 2.3. Statistical Analysis

Demographic data and injury characteristics were reported using descriptive statistics. Nominal data were reported as n (%) and compared using Chi-square or Fisher’s exact tests, depending on expected counts. The normality of continuous data was analyzed visually and statistically via the Shapiro–Wilk and Kolmogorov–Smirnov tests. Non-parametric data were reported as median (25th, 75th percentiles), while parametric data were reported as mean ± standard deviation. To test the primary hypothesis, exposures were first subjected to univariable and then multivariable logistic regression with a manual backward elimination of exposures. Variables were considered significant if *p* < 0.05; however, all variables with *p* < 0.1 during univariable analysis were considered in multivariable modeling and covariates were manually included. Considering potential covariates (i.e., demographics, injury characteristics, and surgical course) as determined by a literature review, multivariable analysis, and Akaike information criterion, TBSA, flame injury, and full-thickness injury produced the most predictive model for the predicted presence of DTp. However, once exposures were entered into the model, the presence of full-thickness injury continually fell out of the model, while the other two covariates remained strongly associated with DTp. It was determined that TBSA and flame injury would need to be included as covariates. The exposure model was verified via repeated measures generalized linear mixed modeling as date of culture within patient as the repeated measure. No differences were observed between the models, so the most parsimonious model was chosen to model exposures. Although the a priori plan for the multicenter study was a Cox-proportional hazard model, survival was analyzed using the Kaplan–Meier method, as the number experiencing the event was few. Cox-proportional hazards will be revisited in the next iteration to better control for covariates that are known to affect mortality after burn injury.

## 3. Results

### 3.1. Patient Demographics

During the one-year study period, 124 patients were admitted to the burn unit and also had a positive culture that resulted in either local or systemic treatment (Figure 1). Thirteen of these patients did not meet inclusion criteria since no treatment was initiated (i.e., initiation of topical or systemic antimicrobials or burn wound excision). Of the 111 eligible patients, 59 were excluded, most commonly for admission for non-burn injuries. The final sample included 65 patients. Fifty-four percent (n = 35) of patients grew a DTp at some point in their hospitalization.

Demographic data including patient comorbidities and HAI RF for the patient population are displayed in Table 1. The average patient in our cohort was a 52-year-old white male. No significant differences were found in age, race, and sex between those who did and did not develop infections with DTp. No significant differences were found between groups in regard to baseline comorbidities with hypertension being the most commonly observed comorbidity (43%, *p* = 0.91), followed by diabetes (32%, *p* = 0.48) and respiratory disorders (15%, *p* = 1). The presence of at least one of the hospital-acquired infection risk factors on admission was fairly common in the cohort (32.2%). The most common risk factor was the presence of a substance abuse history (33.3%), while few patients had end-stage-renal disease (ESRD) requiring dialysis, a history of prior chemotherapy, or intravenous access (IV) on admission.

### 3.2. Injury Characteristics

Injury characteristics, represented in Table 2, did produce some significant differences between those who did and did not develop infections with DTp. An increased TBSA was found in those who developed infections with DTp with a median (25th, 75th percentile) % TBSA of 22.5% (13, 42) versus 9% (3, 12) for those developing infections with non-DTp (*p* < 0.0001). Increased percentages of both partial and full-thickness burns were found to be statistically significant predictors of DTp development (*p* = 0.01 and *p* = 0.008, respectively). Injury mechanism was significantly different between groups (*p* = 0.003). Flame injuries were most commonly associated with DTp development with 88.6% of patients who grew DTp after being injured by flame. Inhalation injury was present in 15% of the cohort, and neither the presence nor grade of inhalation injury was associated with DTp development (0.31 and 0.34, respectively).

### 3.3. Surgical Course

As seen in Table 3, most patients (89.2%) in this cohort were managed surgically, and this was not noted to be different between those who did and did not develop DTp (*p* = 0.23). However, the number of surgeries was found to be statistically associated with DTp development, with a median of three (two, five) surgeries in those who developed DTp and two (one, two) in those who did not (*p* = 0.001). Days to first burn wound excision (BWE) was not noted to be different between groups (*p* = 0.84), but days to final grafting procedure was significantly increased in those developing infections with DTp. Patients who grew DTp had a median day to final grafting procedure of 20 (7, 34) days as opposed to 7 (4, 13) days in those who grew only non-DTp (*p* = 0.003).

### 3.4. Pathogens and Antimicrobial Resistance Patterns

A total of 376 pathogens were isolated from the 65 patients in the cohort. No significant differences were observed in culture source between those who grew DTp and non-DTp (*p* = 0.99). Pathogens were most commonly isolated from wounds (50.5%), followed by blood (23.9%) and the lungs (22.8%). Pathogens were isolated from other sources 2.7% of the time, most often from the urine, though this source was rarely cultured in our study. Here, 57% of pathogens isolated in our cohort were classified as DTp, with the other 43% were classified as non-DTp. Resistance patterns are illustrated in Figure 2. Of the 213 pathogens classified as DTp, 153 (71.8%) of organisms were considered to be MDR, and 46 (21.5%) were classified as XDR. Of DTp, 107 (50.2%) were known or presumed ESBL-producers and 106 (49.8%) were known or presumed AmpC-producers. We observed an overall ESBL and AmpC rate of 28.4% and 28.2%, respectively. Carbapenem-resistance was fairly common, with 17.3% of isolated pathogens exhibiting resistance to these agents. Distributions of organism-specific DTp can also be seen in Figure 2. Of DTp, 62 (29%) were MRSA, 12 (5.6%) difficult-to-treat resistance *Pseudomonas* spp., 9 (4.2%) CRAB, and 26 (12.2%) were *Stenotrophomonas maltophilia.* No vancomycin-resistance was observed in this cohort. The median time to DTp development was found to be 19 (10, 164) days. Infections with MRSA developed most rapidly with a median of 16 (8, 164) days, while DTR-*Pseudomonas* spp. took the longest to develop with a median of 52 (17.75, 88) days. Organisms producing ESBLs or AmpCs, or exhibiting carbapenem resistance, all had a median time to development of 22 days.

### 3.5. Risk Factors for DTp Development

#### 3.5.1. Topical Antimicrobial Exposure

Topical and systemic antimicrobials are commonly utilized in burn patients. Figure 3 illustrates the association between topical exposure and DTp development. We found that 354 (94.1%) of the isolated pathogens were exposed to a topical antimicrobial agent prior to their development and this exposure was associated with a statistically significant increase in DTp development (*p* = 0.0002). Exposures to many topical antimicrobials were found to be statistically significant predictors of DTp development, including bacitracin (*p* < 0.001), Dakins (*p* = 0.003), silver nitrate solution (*p* < 0.001), hypochlorous acid (*p* < 0.001), and other topicals (*p* = 0.0005), largely composed of antifungal agents including nystatin and clotrimazole. Mafenide, silver sulfadiazine (SSD), and mupirocin were not found to be statistically significant predictors of DTp development, though mafenide (i.e., ears) and mupirocin (i.e., most use was a state-mandated routine nasal decolonization before a switch to ethanol-based prophylaxis) are not rarely prescribed in our center. Nasal decolonization with either ethanol-based solutions or mupirocin was not significantly associated with DTp development in either direction (*p* = 0.62). Interestingly, solid silver dressings were found to be associated with a significant decrease in DTp development (*p* = 0.002). During the study period, the only solid silver dressing in use was a nanocrystalline dressing.

#### 3.5.2. Systemic Antimicrobial Exposure

As seen in Figure 4, exposures to many of the tested systemic antimicrobials prior to culture obtainment were also associated with an increased incidence of DTp. In this unadjusted analysis, exposures to non-pseudomonal beta-lactams (e.g., cefazolin, ceftriaxone, or ampicillin-sulbactam) and metronidazole were not found to be associated with DTp development (*p* = 0.95 and 0.66, respectively). In contrast, exposure to all other tested antimicrobials was associated with an increased incidence of DTp development, including: anti-pseudomonal (e.g., cefepime or piperacillin-tazobactam) beta-lactams (*p* < 0.0001), extended-spectrum (e.g., ceftazidime-avibactam, ceftolozane-tazobactam, or meropenem-vaborbactam) beta-lactam beta-lactamase inhibitors (*p* < 0.001), fluoroquinolones (*p* < 0.001), aminoglycosides (*p* < 0.001), antifungal agents (*p* = 0.03), sulfamethoxazole-trimethoprim (*p* < 0.001), carbapenems (*p* < 0.001), and tetracyclines (*p* = 0.01).

#### 3.5.3. Predictors of DTp Development

After identifying which exposure variables were statistically different between those who did and did not develop infections with DTp, we performed a multivariable logistic regression controlling for TBSA and flame injury (see methodology). As shown in Table 4., this analysis identified bacitracin exposure prior to culture obtainment [OR 2.7 (95% C.I. 1.57, 4.65)], silver nitrate solution exposure prior to culture obtainment [OR 1.89 (95% C.I. 1.1, 3.26)], and exposure to anti-pseudomonal beta-lactams prior to culture obtainment [OR 2.6 (95% C.I. 1.46, 4.61)] as the most significant predictors of DTp development.

Results of the individual multivariable logistic regression adjusting for TBSA and flame injury can be seen in Table 5. Six of the seven tested topical antimicrobials retained their significance after controlling for TBSA and flame injury. Bacitracin exposure was the most predictive of these exposures with its use associated with a 3-fold increase in DTp development [OR 3.16 (95% C.I. 1.88, 5.32)]. The other topicals that retained significance were associated with a 1.5- to 2.5-times increased risk of DTp development in the adjusted analysis. Only exposure to Dakin’s was unable to retain significance after controlling TBSA and flame injury. Solid silver dressings were still found to be associated with a decreased incidence of DTp [OR 0.417 (95% C.I. 0.24, 0.72)].

We performed the same individualized analysis for systemic exposures, seen in Table 6, which revealed that seven of the eight previously significant systemic exposures retained their significance irrespective of TBSA and flame injury. The use of fluoroquinolones was the biggest predictor of DTp development [OR 5.8 (95% C.I. 1.31, 25.39)], closely followed by tetracyclines [OR 5.52 (95% C.I. 1.12, 25.15)], though these exposures were much less frequent than exposure to the other tested antimicrobials, which can also be seen in the wide confidence interval. Exposure to aminoglycosides was associated with an over 3-fold increased risk of DTp development [OR 3.45 (95% C.I. 1.82, 6.53)], while exposure to anti-pseudomonal beta-lactams [OR 2.91 (95% C.I. 1.67, 5.06)], MRSA agents [OR 2.73 (95% C.I. 1.58, 4.68)], extended-spectrum beta-lactam beta-lactamase inhibitors [OR 2.25 (95% C.I. 1.26, 4.03)], and sulfamethoxazole-trimethoprim [OR 2.14 (95% C.I. 1.21, 3.79)] were associated with a 2–3-times increased risk of DTp development. Exposure to non-pseudomonal beta-lactams gained significance after controlling for TBSA and flame injury [OR 1.67 (95% C.I. 1.04, 2.69)]. Exposure to antifungal agents was not found to be associated with DTp development after controlling for TBSA and flame, though antifungal agents were uncommon.

### 3.6. Impact of DTp Development on Mortality

We observed a mortality rate of 6% in our cohort, with all patient deaths being those who developed infections with DTp. As there were only four deaths in our study, we were unable to perform a Cox-proportional hazard analysis. However, a Kaplan–Meier plot (Figure 5) stratified by DTp development revealed a clear divergence in mortality between those who did and did not develop infections with DTp (Log rank *p* = 0.0583).

## 4. Discussion

Antimicrobial resistance is a pervasive problem for burn centers worldwide. To our knowledge, this is the first comprehensive analysis to identify both covariates and exposures for the predictive modeling of DTp. It is likely that the incidence of DTp will be significantly influenced by a number of different institution-specific factors including the center’s microbiological ecosystem, wound care and infection prevention protocols, surgical management, and the varied use of antimicrobials, among others. As such, the data from this pilot analysis are incredibly informative and demonstrate feasibility for the next multicenter iteration. The next step is incredibly important for the future care of patients following burn injury.

To differentiate colonization from true infection, we defined infection as a culture result that led to the initiation of treatment (i.e., initiation of topical or systemic antimicrobials or burn wound excision). Our data support the notion of severe flame injury to producing a near sterile wound that will be rapidly recolonized with pathogens from the immediate environment (perhaps at the scene of extraction or in the forthcoming healthcare environment). In the hospital setting, this will often mean wound colonization with drug-resistant organisms, though these colonizations do not always lead to clinically significant infectious complications. A study by Gallaher et al. found that 70% of burn patients had wounds colonized by MDR organisms at some point during hospitalization [34]. The inclusion of surveillance cultures in our study would have likely led to an overestimation in the incidence of DTp. We found that over half of the patients included grew a DTp at some point in their hospital stay, with 57% of all pathogens being classified as DTp. The incidence of DTp among burn centers has been highly variable, with reported incidence between 11% and 79% [4,30,34,35,36,37,38,39]. This variability reveals an opportunity to identify which factors play a role in the development of these resistant organisms, and potentially intervene before their development occurs. Previous studies have illustrated that burn wound infections with Gram-positive organisms are prevalent early in hospitalization, while pneumonia and bacteremia with Gram-negative organisms become more prevalent as length of stay increases [32,40,41]. A study by Van Duin et al. evaluating 1788 bacterial isolates found the median time to development of MRSA, ESBL-enterobacterales, CRE, and MDR-*Pseudomonas* spp. to be 11.5, 52, 59, and 52 days, respectively [41]. Our study had similar findings, with median time to MRSA and DTR- *Pseudomonas* spp. being 16 and 52 days, respectively. However, we did find a significantly decreased time to development of ESBL-producing and carbapenem-resistant organisms, with a median time to development of 22 days. These results underscore the importance of preserving our broadest antimicrobial agents for infections occurring later in the hospital length of stay, though the exact timing requires further elucidation.

Several well-established risk factors exist for the development of DTp in burn patients. However, many of these risk factors are non-modifiable, including flame injury, increased TBSA burned, and the presence of inhalation injury [33,42,43]. We were unable to confirm inhalation injury as a significant predictor of DTp development, though only 10 patients in the study had these injuries. Inhalation injury is diagnosed solely through bronchoscopy in our center, giving insight to the incidence. However, increased TBSA and flame injuries were noted to be significant predictors of DTp development. Early burn wound excision (i.e., within 7 days of injury) has been shown to be one of the most important factors in preventing infectious complications following burn injury, as necrotic tissue is a rich medium for pathogen growth [22,23]. Delayed wound excision beyond this time point has been associated with an increased incidence of invasive wound infections and sepsis [22]. Time to first burn wound excision was within 4 days of admission for all patients in the study, and no significant differences were found in terms of DTp development. An increased number of surgical procedures and an increased time to final grafting procedure were both independently associated with an increased incidence of DTp development, highlighting the importance of prompt wound closure. These factors may represent inadequate, or delayed, source control, leading to increased exposure to topical or systemic antimicrobial agents.

Both topical and systemic antimicrobials are often overused in the burn population, as differentiating burn sepsis from the hypermetabolic response to burn injury can be difficult. A study published by Hill et al. determined that 95% of patients meet systemic inflammatory response syndrome (SIRS) criteria for the entirety of their hospital stay. Utilizing SIRS criteria as the sole criteria for initiating antibiotics will lead to multiple exposures and prolonged courses of antimicrobial agents [44]. A retrospective analysis evaluating the incidence of respiratory infections in a burn intensive care unit (ICU) compared to the hospitals’ other seven ICUs found a significant increase in ventilator-acquired pneumonia with MDR Gram-negative bacilli in burn patients compared to other critically ill patients [45]. Additionally, a study by Gong et al. found increased rates of MDR-*Acinetobacter baumannii, Pseudomonas aeruginosa*, and *Klebsiella pneumoniae* between burn patients located in the burn ICU as compared to the common burn ward [24]. The increased rates of MDR organisms observed in burn patients are likely related to the overuse of antimicrobial agents in this population, and may represent a difference in environmental colonization between the units. The results of our study echo the current literature, which illustrates the increased prevalence of DTp following exposures to systemic antimicrobials [11,12,13,14,15,17]. We controlled for known confounders and found that exposure to 8 of the 11 tested antimicrobial classes led to an increased incidence of DTp. Even exposure to narrow-spectrum agents, such as non-pseudomonal beta-lactams, were associated with an increased rate of DTp development after controlling for TBSA and flame injury, which illustrates the importance of stringent antimicrobial stewardship practices even when using narrow-spectrum agents.

The use of topical antimicrobials is often viewed as a benign intervention and overlooked as a cause of antimicrobial resistance. Topical agents are often utilized in burn patients and have been associated with both increased rates of antimicrobial resistance and impaired wound healing [16,18,19]. If circumstances allow for prompt and frequent attention to burn wounds, appropriate debridement and wound care with soap and clean water is preferred to topical antimicrobials for the prevention of wound infection [33]. The use of topical antimicrobial agents is best preserved for when adequate source control is delayed. However, our study illustrates a pervasive problem in burn care, as 94% of pathogens isolated in our study were exposed to at least one topical antimicrobial agent. Over half of the tested antimicrobials in this study were associated with an increased incidence of DTp after adjustment. In fact, exposure to two topical antimicrobials, bacitracin and silver nitrate solution, were identified as the biggest predictors for DTp development alongside exposure to anti-pseudomonal beta-lactams. The judicious use of these agents is vital in the prevention of DTp development.

Numerous studies have cited an increase in mortality associated with the development of DTp in general hospitalized or ICU patients [2,10,27,28]. However, this relationship is less well established in burn patients. A 2021 study of 58 burn patients with a TBSA of 15% or greater found a mortality rate of 18% in patients who developed an infection with an MDR organisms as compared to 16% of patients with non-MDR infections (*p* = 1) [30]. Another study by Theodorou et al. found that among 87 burn patients with *Pseudomonas aeruginosa* bacteremia, there was no difference in mortality for those who developed infections with MDR strains as compared to those who did not (*p* = 0.897) [31]. There were only four deaths in the present study, which may have affected our ability to detect statistical differences between groups; however, a clear divergence in mortality was observed, as all patients who died developed infections with DTp. While mortality was low, all of the deaths were in patients that were being treated for DTp. A multicenter analysis will be critical for this next analysis, as we can control for a possible center effect.

This study is not without its limitations. This was a retrospective, single-center study, which may affect the generalizability of the results. We had a small sample size of 65 patients, which may have affected our ability to detect statistical differences between groups regarding the univariate analysis of DTp incidence and mortality. However, the sample was robust enough to perform regression analysis with a repeated measures design, as nearly 400 pathogens were included (~5.8 pathogens/patient). Additionally, our center rarely cultures urine, which may have led to an underestimation of pathogen burden. Lastly, the collection of topical exposure was difficult to obtain, as topical antimicrobials were often ordered but not always charted appropriately in the electronic medical record. As such, for the sake of the study of topical exposure, topicals were considered as either documented in the medication administration record as administered, or documented as applied in the nurse or prescriber’s wound care or progress notes.

This study revealed a high rate (57%) of DTp development, and identified many risk factors for the development of DTp in patients with burn injuries. The findings support existing primary literature on injury characteristics with high risk, such as TBSA and flame injury, as well as new data associating the use of many topical and systemic antimicrobials with resistance risk. The probability of survival after acquiring a DTp diverged but was not statistically significant in this sample. The results of this study highlight the need for a multicenter study to more sufficiently explore the variation in microbial ecosystems and risk factors associated with the development of DTp in burn patients between centers. As mentioned previously, clinical practices vary significantly from center to center, and likely impact the development of DTp. We plan to use our pilot results to inform our data collection in the multicenter study to evaluate the impacts these varied practices have on DTp development. For burn patients who survive the initial resuscitation, sepsis is the leading cause of death [33]. This fact alone highlights the importance of identifying specific risk factors for DTp development that can be universally applied to enable the better care of our patients and inform the next phase of therapeutic development.

## Figures and Tables

**Figure 1 pathogens-13-00628-f001:**
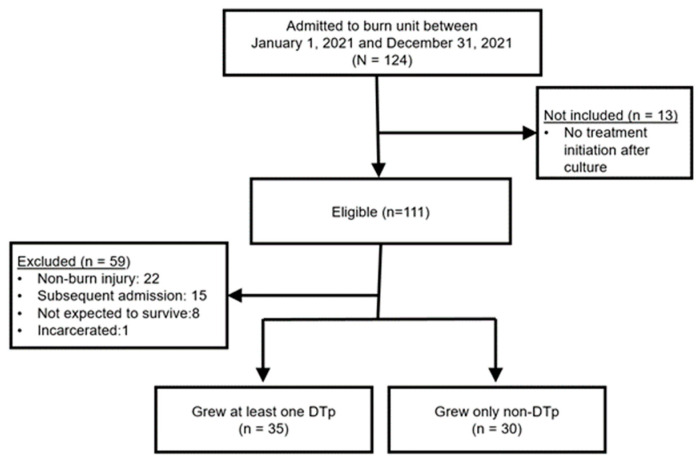
Flow diagram of patient screening and final cohort. N = patients screened; n = final sample of patients.

**Figure 2 pathogens-13-00628-f002:**
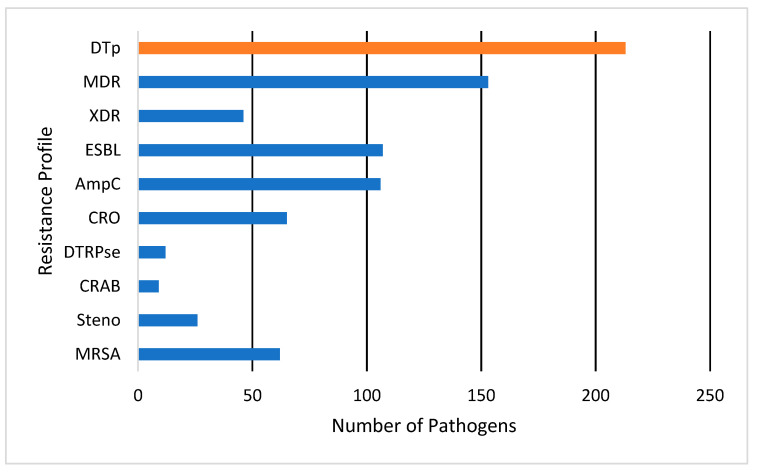
Antimicrobial resistance profiles for pathogens classified as DTp. DTp is a composite (show in orange) of the listed individual antimicrobial resistance profiles (shown in blue). Pathogens could meet criteria for multiple resistance categories. MDR, multi-drug resistant; XDR, extensively drug resistant; ESBL, extended-spectrum beta-lactamase; AmpC, ampicillin-resistance gene group C; CRO, carbapenem-resistant organism; DTRPse, difficult-to-treat resistance *Pseudomonas* spp.; CRAB, carbapenem-resistant *Acinetobacter baumannii*; Steno, *Stenotrophomonas maltophilia;* MRSA, methicillin-resistant *Staphylococcus aureus*.

**Figure 3 pathogens-13-00628-f003:**
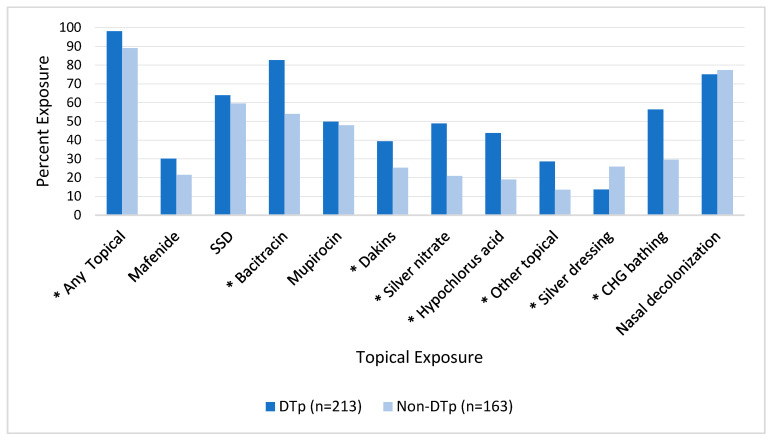
Impact of exposure to topical antimicrobials prior to culture obtainment on DTp development. * denotes *p* < 0.05. SSD, silver sulfadiazine; CHG, chlorohexidine gluconate.

**Figure 4 pathogens-13-00628-f004:**
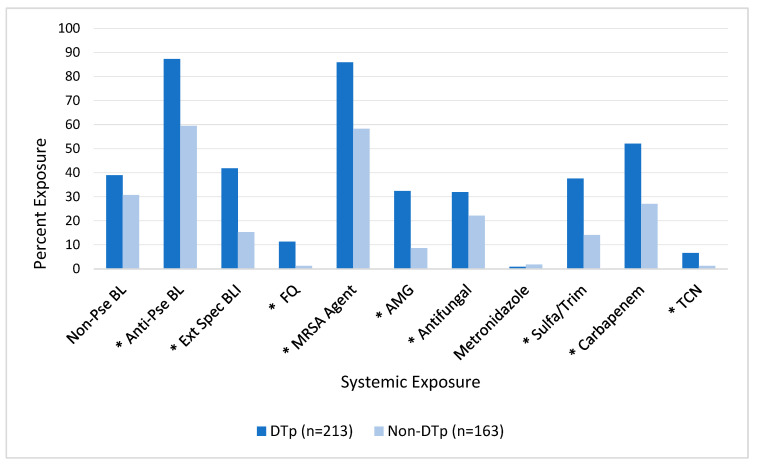
Impact of exposure to systemic antimicrobials prior to culture obtainment on DTp development. * Denotes *p* < 0.05. Non-Pse BL, non-pseudomonal beta-lactam; Anti-Pse BL, anti-pseudomonal beta-lactam; Ext Spec BLI, extended-spectrum beta-lactam beta-lactamase inhibitor; FQ, fluoroquinolone; MRSA, methicillin-resistant *Staphylococcus aureus*; AMG, aminoglycoside; Sulfa/Trim, sulfamethoxazole-trimethoprim; TCN, tetracycline.

**Figure 5 pathogens-13-00628-f005:**
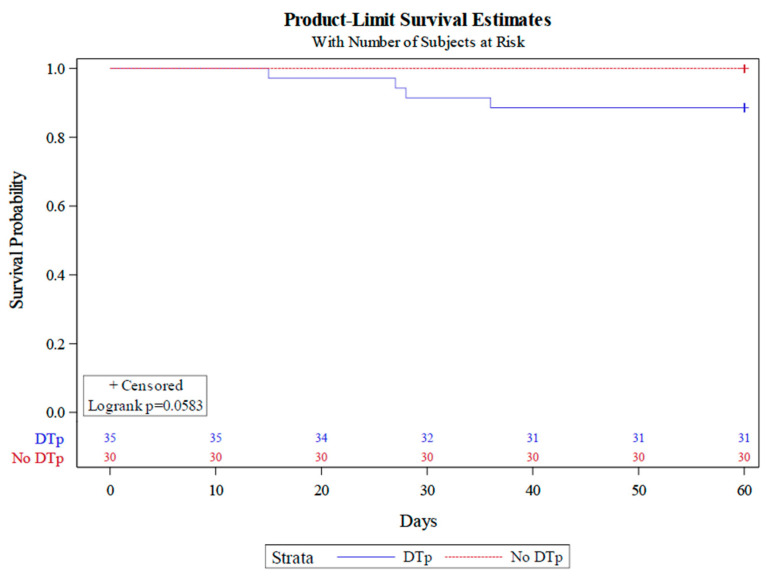
Kaplan–Meier analysis stratified by DTp development.

**Table 1 pathogens-13-00628-t001:** Baseline characteristics between those who did and did not develop DTp.

	DTp (n = 35)	Non-DTp (n = 30)	*p*-Value
Age, years ^a^	52.1 ± 16.6	53.2 ± 17.6	0.79
Male ^b^	24 (68.6)	23 (76.7)	0.46
Caucasian ^b,c^	23 (65.7)	17 (56.7)	0.45
DM ^b^	10 (28.6)	11 (26.7)	0.48
ESRD ^b^	1 (2.9)	1 (3.3)	0.91
HTN ^b^	15 (42.9)	13 (43.3)	0.96
HF ^b^	3 (8.6)	1 (3.3)	0.38
HLD/CAD ^b^	6 (17.1)	3 (10)	0.40
AF ^b^	2 (5.7)	0	0.49
Stroke ^b^	4 (11.4)	1 (3.3)	0.36
COPD/Asthma ^b^	5 (14.3)	5 (16.7)	1
Psychiatric Disorder ^b^	6 (17.1)	2 (6.7)	0.26
HAI RF ^b,d,e^	13 (37.1)	8 (26.7)	0.27
IV access ^b^	0	1 (3.3)	0.46
History of chemotherapy ^b^	0	1 (3.3)	0.46
Substance abuse history ^b^	13 (37.1)	7 (23.3)	0.23

DM, diabetes mellitus; ESRD, end-stage renal disease; HTN, hypertension; HF, heart failure; HLD, hyperlipidemia; CAD, coronary artery disease; AF, atrial fibrillation; COPD, chronic obstructive pulmonary disease; HAI RF, hospital-acquired infection risk factor; IV, intravenous; ^a^ Mean ± SD, ^b^ n (%), ^c^ only 1 Hispanic, remaining non-white were black, ^d^ Will not add up to 100% as patients could have more than one risk-factor, ^e^ patients admitted from nursing home or healthcare exposure in last 90-days.

**Table 2 pathogens-13-00628-t002:** Injury characteristics.

	DTp (n = 35)	Non-DTp (n = 30)	*p*-Value
TBSA ^a^	22.5 (13, 42)	9 (3, 12)	<0.0001
PT (%) ^a^	9 (1.5, 19)	2.8 (0, 7.8)	0.01
FT (%) ^a^	9.3 (2, 18.5)	1.9 (0, 8.5)	0.008
Injury mechanism ^b^			0.003
Flame	31 (88.6)	15 (50)	
Scald	1 (2.9)	7 (23.3)	
Hot contact	2 (5.7)	4 (13.3)	
Other	1 (2.9)	4 (13.3)	
Presence of inhalation injury ^c^	3	7	0.31
Grade of injury ^b^			0.34
1	0	1 (2.9)	
2	1 (3.3)	5 (14.3)	
3	0	0	
4	2 (6.7)	1 (2.9)	

TBSA, total body surface area; PT, partial thickness; FT, full thickness; ^a^ median (25th, 75th percentile), ^b^ n (%), ^c^ n.

**Table 3 pathogens-13-00628-t003:** Surgical course.

	DTp (n = 35)	Non-DTp (n = 30)	*p*-Value
Surgery ^a^	33 (94.3)	25 (83.3)	0.23
Number of surgeries ^b^	3 (2, 5)	2 (1, 2)	0.001
Days to first BWE ^b^	2 (1, 3)	2 (1, 4)	0.84
Days to final graft ^b^	20 (7, 34)	7 (4, 13)	0.003

BWE, burn wound excision; ^a^ n (%), ^b^ median (25th, 75th percentile).

**Table 4 pathogens-13-00628-t004:** Multivariable logistic regression of DTp risk factors controlling for TBSA and flame injury.

	OR	95% CI	*p*-Value
TBSA	1.007	0.99, 1.02	0.38
Flame injury	4.5	1.58, 12.76	0.005
Bacitracin exposure	2.7	1.57, 4.65	0.003
Silver nitrate exposure	1.89	1.1, 3.26	0.02
AP beta-lactam exposure	2.6	1.46, 4.61	0.001

AP, anti-pseudomonal; CI, confidence interval; OR, odds ratio; TBSA, total body surface area.

**Table 5 pathogens-13-00628-t005:** Results of the individual multivariable logistic regressions for topical exposures found to be statistically significant.

Exposure	OR	95% CI
Bacitracin	3.16	1.88, 5.32
Flame	5.00	1.84, 13.54
TBSA	1.02	1, 1.03
Silver nitrate solution	2.58	1.54, 4.32
Flame	4.29	1.6, 11.5
TBSA	1.01	1, 1.03
Hypochlorous acid	2.44	1.45, 4.11
Flame	4.15	1.55, 11.08
TBSA	1.02	1, 1.03
Other topical	1.99	1.14, 3.46
Flame	3.16	1.19, 8.38
TBSA	1.02	1.01, 1.04
Solid silver dressing	0.42	0.24, 0.72
Flame	3.88	1.46, 10.28
TBSA	1.02	1.01, 1.04
CHG bathing	1.81	1.1, 2.97
Flame	3.56	1.35, 9.4
TBSA	1.02	1, 1.03

CHG, chlorohexidine gluconate; CI, confidence interval; OR, odds ratio; TBSA, total body surface area.

**Table 6 pathogens-13-00628-t006:** Results of the individual multivariable logistic regression for systemic exposures found to be statistically significant.

Exposure	OR	95% CI
Non-pse BL	1.67	1.04, 2.69
Flame	3.55	1.34, 9.39
TBSA	1.03	1.01, 1.04
Anti-pse BL	2.91	1.67, 5.06
Flame	2.91	1.08, 7.84
TBSA	1.02	1, 1.03
FQ	5.8	1.32, 25.39
Flame	3.69	1.4, 9.7
TBSA	1.02	1.01, 1.03
MRSA Agent	2.73	1.58, 4.68
Flame	30.3	1.4, 9.7
TBSA	1.02	1, 1.03
AMG	3.46	1.82, 6.53
Flame	3.26	1.12, 8.59
TBSA	1.02	1.01, 1.03
Sulfa/trim	2.14	1.21, 3.79
Flame	3.57	1.35, 9.38
TBSA	1.02	1, 1.03
ESBLI	2.25	1.26, 4.03
Flame	3.71	1.41, 9.74
TBSA	1.01	1, 1.03
TCN	5.52	1.21, 25.15
Flame	3.32	1.25, 8.77
TBSA	1.02	1.01, 1.04

AMG, aminoglycoside; Anti-Pse BL, anti-pseudomonal beta-lactam; CI, confidence interval; ESBLI, extended spectrum beta-lactam beta-lactamase inhibitor; FQ, fluoroquinolone; MRSA, methicillin-resistant *Staphylococcus aureus*; Non-Pse BL, non-pseudomonal beta-lactam; OR, odds ratio; Sulfa/Trim, sulfamethoxazole-trimethoprim; TCN, tetracycline.

## Data Availability

Dataset available on request from the authors.

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
