# Peer review of "A Pilot Analysis for a Multicentric, Retrospective Study on Biodiversity and Difficult-to-Treat Pathogens in Burn Centers across the United States (MICROBE)"

_pathogens, 2024, doi:10.3390/pathogens13080628_

Round 1
Reviewer 1 Report
Comments and Suggestions for Authors
This manuscript discusses the infection situation of drug-resistant bacteria in burn patients. It starts with an analysis of cases selected from hospitalized patients that meet statistical requirements, studying the bacterial flora, drug resistance, and prognosis of these infections. After reviewing the manuscript, I have a few questions to discuss with the author:
- The manuscript classifies the types of bacteria causing infections, but the author has not provided the methods for isolating and identifying these strains. Please include this information.
- Please provide the methods used for identifying drug resistance. Similar to question 1, although the author mentioned using BD Phoenix™ for strain identification and antibiotic susceptibility testing, the methods for sample processing need to be included.
Author Response
Thank you very much for taking the time to review this manuscript. We appreciate your thoughtful comments to improve our manuscript. Please find the detailed responses below and the corrections highlighted in the re-submitted files.
Reviewer Comments:
The manuscript classifies the types of bacteria causing infections, but the author has not provided the methods for isolating and identifying these strains. Please include this information.
Thank you for this comment. More information regarding the isolation and identification of microbial organisms has been added in lines 146-171.
Please provide the methods used for identifying drug resistance. Similar to question 1, although the author mentioned using BD Phoenix™ for strain identification and antibiotic susceptibility testing, the methods for sample processing need to be included.
We have updated the methods to include more information regarding this process in lines 146-171.
Reviewer 2 Report
Comments and Suggestions for Authors
This is a well described and important study that evaluates the incidence of predefined "difficult to treat" pathogens among burn patients. Methods are well explained, analysis and results presentation are vigorous, and discussion is complete, with comparisons, clearly presented findings and conclusions. Limitations are also acknowledged, as this is a retrospective single-center analysis and limited subject numbers may have hindered the results of statistical analysis.
I have minor comments:
Please provide additional information in the author affiliations.
Lines 121-125: please define which antimicrobials belong to each group.
Line 167: "As you can see" please rephrase.
Line 402 "All patients in the present study were initially excised" please rephrase.
Author Response
Thank you very much for taking the time to review this manuscript. We appreciate your thoughtful comments to improve our manuscript. Please find the detailed responses below and the corrections highlighted in the re-submitted files.
Reviewer Comments:
Please provide additional information in the author affiliations.
Affiliations have been updated
Lines 121-125: please define which antimicrobials belong to each group.
Thank you for this suggestion. This change has been made in the manuscript.
Line 167: "As you can see" please rephrase.
Thank you for catching this mistake. This has been rephrased in manuscript.
Line 402 "All patients in the present study were initially excised" please rephrase.
Thank you for this suggestion. This has been rephrased in manuscript.
Reviewer 3 Report
Comments and Suggestions for Authors
I suggest the title to be: Multicentric retrospective study on biodiversity and difficult-to-treat pathogens in burn centres in United States
16: Define all the abbreviations in the abstract and remove abbreviations
The methodology for the isolation and identification of the organisms were not described in the abstract. The conclusion of the abstract should be stated.
31: risk for or risk of?
34: difficult to treat should be difficult-to-treat
72,11,186, 383, 419: What is the full meaning of the following first time used: IRB, CHG, IV, ESBL-E, ICU
This study involved data from burn patients but no informed consent was obtained from the patients prior to the usage of their data
94: Culture results were utilized in the study, but the methods used in sample (wound swabs/skin scrapings) collection, and isolation of the organisms were not described neither was reference(s) of previous study(ies) that might have reported the methods cited. Similarly, organisms were identified to species level, but the identification method(s) used was not mentioned
100,110: vancomycin resistant should be vancomycin-resistant
102: carbapenem resistant should be carbapenem-resistant
132: For reproducibility, mention the method used to screen the Enterobacteriaceae for KPC enzymes. What method was employed italicize and check the spelling of Klebsiella pneumoniae
139: Cite and list in the reference the CLSI breakpoints used for the classification of isolates tested by disc diffusion. Also mention the antimicrobial agents that were tested by disc diffusion. The bacterial strain used for quality control in the sensitivity test was not mentioned neither a previous paper that reported the testing of the isolates cited.
149: Stated that “Significant variables were considered p < 0.1…” but it was not mentioned that for effect of topical and systemic antibiotic exposure on the development of difficult-to-treat pathogens, the probability was set at p < 0.05 as shown in figures 3 and 4
167: “As you can see” should not be used in a scientific paper
179: This statement should be in the methodology “Baseline comorbidities were assessed between those who did and did not develop infections with DTp”
180-182: What percentage/proportion of the sampled individuals had each of the comorbidities mentioned (also in 185-186). What was the probability that showed there was no significantly statistical association (also in line 269 and 337)
Table 1, 2, 3, 4, 5 are not self-explanatory as there were unexplained abbreviations. The 95% Confidence interval should be written as a range. Figure 2 lacked title on the vertical axis. The chart title “antimicrobial resistance” in the chart area as well as “Topical exposures” in figure 3, should be removed. “difficult-to-treat resistance Pseudomonas spp.” under figure 2 should be “resistant”. Figures 3 and 4 lacked horizontal axis title
There should be result indicating the isolation rates of the different types of organisms (ESBL-producing Enterobacteriaceae: E. coli, Slmonella, Klebsiella etc), Sternotrophomonas, etc
230,237,366: Check spelling errors, 241: do not italicize spp. and throughout the manuscript; 268: “clomitrazole”
228-233: Is a repetition of the criteria for classifying isolates as difficult-to-treat pathogens in the methodology lines 97-112. Similarly, lines 258-260 is a repetition of lines114-125. Lines 310-311 is equally a repetition of the methodology
There is a need for clarification of contradictory statements. Risk factors for DTp development in lines 274-275 and 317 showed that “silver dressings were found to be associated with a significant decrease in DTp development (p=0.002).” whereas predictors of DTp development in lines 303-304 revealed that “silver nitrate exposure prior to culture obtainment [OR 1.89 (95% C.I. 1.1, 3.26)],”
385,387: In discussion, where did 16, 22 and 52 days emanate from? The results of these median times for development of MDR Pseudomonas were not mentioned in the result section.
425: There was no mention that exposure to 8 of the 11 tested antimicrobial classes led to an increased incidence of DTp in the result section, the data must be presented for it to be discussed
426-428: Stated that even exposure to narrow spectrum led to development of DTp. The discussion should about the implication of this finding and whether there was statistically significant difference with the rate broad spectrum and marrow-spectrum induce multidrug resistance
432-433: Incomprehensible sentence
There is no comprehensive conclusion of the study which should be directed to the findings of the study. Recommendation or significance of the study is not the same as the conclusion.
Comments on the Quality of English Language
Minor English editing required
Author Response
Thank you very much for taking the time to review this manuscript. We appreciate your thoughtful comments to improve our manuscript. Please find the detailed responses below and the corrections highlighted in the re-submitted files.
Reviewer Comments:
I suggest the title to be: Multicentric retrospective study on biodiversity and difficult-to-treat pathogens in burn centres in United States
We have updated the title. Thank you for this suggestion.
16: Define all the abbreviations in the abstract and remove abbreviations –
Has been updated in the manuscript.
The methodology for the isolation and identification of the organisms were not described in the abstract. The conclusion of the abstract should be stated. –
Unfortunately, due to word limits we are unable to include a full methodology in the abstract regarding isolation and identification. However, methodology was expanded according to suggestions from referee 2 and 3. The abstract conclusion was updated.
31: risk for or risk of? –
Has been updated in the manuscript to “risk of”
34: difficult to treat should be difficult-to-treat –
This change has been made throughout the manuscript.
72,11,186, 383, 419: What is the full meaning of the following first time used: IRB, CHG, IV, ESBL-E, ICU –
Thank you for drawing our attention to the missing detail. Definitions have been added to abbreviations upon first occurrence
This study involved data from burn patients but no informed consent was obtained from the patients prior to the usage of their data –
Yes this is correct. Necessity of informed consent was waived by two Institutional Review Boards. Section 2.1 was updated to make this more clear.
94: Culture results were utilized in the study, but the methods used in sample (wound swabs/skin scrapings) collection, and isolation of the organisms were not described neither was reference(s) of previous study(ies) that might have reported the methods cited. Similarly, organisms were identified to species level, but the identification method(s) used was not mentioned –
More information regarding sample collection was added to lines 99-101. Organism identification and susceptibility testing using the BD Phoenix™ automated system or MALDI TOF BioTyper are discussed in lines 140-171.
100,110: vancomycin resistant should be vancomycin-resistant –
This has been changed throughout manuscript.
102: carbapenem resistant should be carbapenem-resistant
This has been changed throughout manuscript.
132: For reproducibility, mention the method used to screen the Enterobacteriaceae for KPC enzymes. What method was employed italicize and check the spelling of Klebsiella pneumonia –
Thank you for this recommendation. This sentence was included in error as our institution does not locally perform genetic testing. The Phoenix system uses a CPO Detect Panel, but specific genetic testing requires samples to be sent out to consulting laboratories and are per physician request. The sentence has been updated and Klebsiella pneumoniae has been italicized and spelling verified.
139: Cite and list in the reference the CLSI breakpoints used for the classification of isolates tested by disc diffusion. Also mention the antimicrobial agents that were tested by disc diffusion. The bacterial strain used for quality control in the sensitivity test was not mentioned neither a previous paper that reported the testing of the isolates cited.
Thank you for this comment. We have added information regarding quality control. Disk diffusion was rare and was isolated to the use of ceftazadime-avibactam, meropenem-vaborbactam, and ceftolozane-tazobactam for the treatment of MDR Acinetobacter spp., Enterobacterales spp., and Pseudomonas spp. This has been updated in the text.
149: Stated that “Significant variables were considered p < 0.1…” but it was not mentioned that for effect of topical and systemic antibiotic exposure on the development of difficult-to-treat pathogens, the probability was set at p < 0.05 as shown in figures 3 and 4.
Clarification was added to Section 2.3.
167: “As you can see” should not be used in a scientific paper –
This has been rephrased in line 167.
179: This statement should be in the methodology “Baseline comorbidities were assessed between those who did and did not develop infections with DTp” –
We agree with this. These statement has been removed from results section and listed in the methods.
180-182: What percentage/proportion of the sampled individuals had each of the comorbidities mentioned (also in 185-186). What was the probability that showed there was no significantly statistical association (also in line 269 and 337) –
Percentages and p values added to text and are listed in Table 1.
Table 1, 2, 3, 4, 5 are not self-explanatory as there were unexplained abbreviations. The 95% Confidence interval should be written as a range. Figure 2 lacked title on the vertical axis. The chart title “antimicrobial resistance” in the chart area as well as “Topical exposures” in figure 3, should be removed. “difficult-to-treat resistance Pseudomonas spp.” under figure 2 should be “resistant”. Figures 3 and 4 lacked horizontal axis title –
Thank you for the recommendation. We have checked that all table abbreviations were defined, added vertical axis title on Fig. 2, removed chart titles on fig. 2,3, and 4 and updated horizontal and vertical axis titles, difficult-to treat resistance Pseudomonas is a term defined by the IDSA as “not susceptible to at least one antibiotic in at least three antibiotic classes for which P. aeruginosa susceptibility is generally expected: penicillins, cephalosporins, fluoroquinolones, aminoglycosides, and carbapenems”, definition added to methods section.
There should be result indicating the isolation rates of the different types of organisms (ESBL-producing Enterobacteriaceae: E. coli, Slmonella, Klebsiella etc), Sternotrophomonas, etc –
During manual data collection, all ESBL-producing Enterobacteriaceae were classified as such, so individual species data is not available. Isolation rates of other organisms including Stenotrophomonas are listed in the text (lines 256-262) and can be seen in Figure 2.
230,237,366: Check spelling errors, 241: do not italicize spp.  and throughout the manuscript; 268: “clomitrazole” –
The spelling has been corrected, Spp. Italics removed throughout the manuscript.
228-233: Is a repetition of the criteria for classifying isolates as difficult-to-treat pathogens in the methodology lines 97-112. Similarly, lines 258-260 is a repetition of lines114-125. Lines 310-311 is equally a repetition of the methodology –
All duplicate methodology mentioned rephrased or removed from results section.
There is a need for clarification of contradictory statements. Risk factors for DTp development in lines 274-275 and 317 showed that “silver dressings were found to be associated with a significant decrease in DTp development (p=0.002).”  whereas predictors of DTp development in lines 303-304 revealed that “silver nitrate exposure prior to culture obtainment [OR 1.89 (95% C.I. 1.1, 3.26)],” –
Thank you for drawing our attention to this detail. We agree the phrasing is confusing. Solid silver dressings (nanocrystalline) and liquid (silver nitrate solution) were used and considered individually. This distinction has been made throughout the manuscript.
385,387: In discussion, where did 16, 22 and 52 days emanate from? The results of these median times for development of MDR Pseudomonas were not mentioned in the result section. –
Thank you for this question. Days to developing DTp is discussed beginning at line 259 of the results in section 3.4.
425: There was no mention that exposure to 8 of the 11 tested antimicrobial classes led to an increased incidence of DTp in the result section, the data must be presented for it to be discussed –
Thank you for this suggestion. The results were depicted in Figure 3 and presented in section 3.5.1.
426-428: Stated that even exposure to narrow spectrum led to development of DTp. The discussion should about the implication of this finding and whether there was statistically significant difference with the rate broad spectrum and marrow-spectrum induce multidrug resistance –
Thank you for the suggestion. Implications were added beginning at line 440.
432-433: Incomprehensible sentence –
Thanks for the comment. The sentence has been modified (line 448).
There is no comprehensive conclusion of the study which should be directed to the findings of the study. Recommendation or significance of the study is not the same as the conclusion. –
A more clear conclusion was added to the final paragraph, highlighting the results of the study.